# Conditional Panoramic Image Generation via Masked Autoregressive Modeling

**Chaoyang Wang**[1]  **Xiangtai Li**[1]  **Lu Qi**[2*]  **Xiaofan Lin**[2]
**Jinbin Bai**[3]  **Qianyu Zhou**[4]  **Yunhai Tong**[1]
[1]School of Intelligence Science and Technology, Peking University
[2]Insta360 Research  [3]National University of Singapore  [4]The University of Tokyo
cywang@stu.pku.edu.cn, qqlu1992@gmail.com

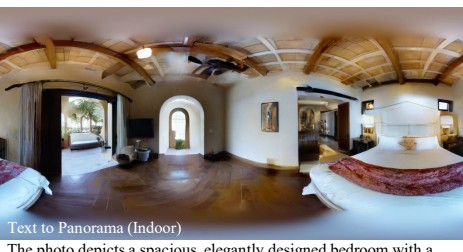

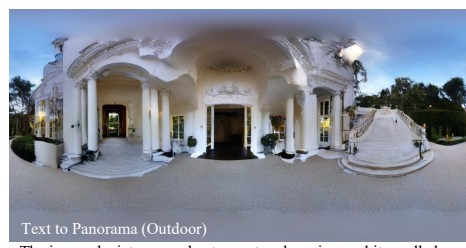

Text to Panorama (Indoor)

The photo depicts a spacious, elegantly designed bedroom with a large bed, wooden beams on the ceiling, and a view of a balcony through arched windows.

Text to Panorama (Outdoor)

The image depicts a grand entrance to a luxurious, white-walled building with arched doorways, a staircase, and a well-maintained garden area.

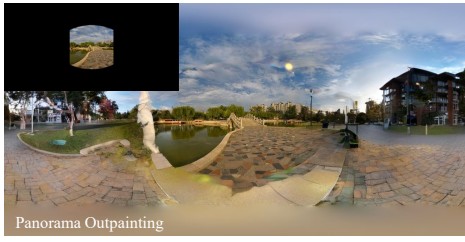

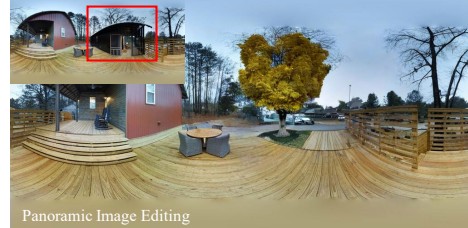

Panorama Outpainting

A park with a pond and buildings.

Panoramic Image Editing

A yellow tree in the yard.

Figure 1: **Generated samples from Panoramic AutoRegressive (PAR) Model**. PAR unifies several conditional panoramic image generation tasks, including text-to-panorama, panorama outpainting, and panoramic image editing.

## Abstract

Recent progress in panoramic image generation has underscored two critical limitations in existing approaches. First, most methods are built upon diffusion models, which are inherently ill-suited for equirectangular projection (ERP) panoramas due to the violation of the identically and independently distributed (*i.i.d.*) Gaussian noise assumption caused by their spherical mapping. Second, these methods often treat text-conditioned generation (text-to-panorama) and image-conditioned generation (panorama outpainting) as separate tasks, relying on distinct architectures and task-specific data. In this work, we propose a unified framework, Panoramic AutoRegressive model (PAR), which leverages masked autoregressive modeling to address these challenges. PAR avoids the *i.i.d.* assumption constraint and integrates text and image conditioning into a cohesive architecture, enabling seamless generation across tasks. To address the inherent discontinuity in existing generative models, we introduce circular padding

*Corresponding author

39th Conference on Neural Information Processing Systems (NeurIPS 2025).

to enhance spatial coherence and propose a consistency alignment strategy to improve the generation quality. Extensive experiments demonstrate competitive performance in text-to-image generation and panorama outpainting tasks while showcasing promising scalability and generalization capabilities. Project Page: https://wang-chaoyang.github.io/project/par.

# 1 Introduction

Panoramic image generation[2, 12, 13, 18, 19] has recently emerged as a pivotal research direction in computer vision, driven by its capability to capture immersive 360-degree horizontal and 180-degree vertical fields of view (FoV). This unique visual representation closely mimics human panoramic perception, enabling transformative applications across VR/AR [34, 79], autonomous driving [44], visual navigation [6, 25], and robotics [45, 60].

Despite some progress in recent years, existing panorama generation methods [18, 56, 57, 74, 70, 27, 36, 62, 77, 65, 64, 76] predominantly rely on diffusion-based base models and face two significant problems: task separation and inherent limitations of the modeling approach. First, different panoramic generation tasks are highly bound to specific foundation models. For instance, most text-to-panorama models [18, 56, 57, 74, 70] are based on Stable Diffusion (SD) [48, 43]. In contrast, panorama outpainting models [27, 36, 62] are fine-tuned on SD-inpainting variants, resulting in task-specific pipelines that lack flexibility. Second, diffusion-based models struggle to handle the unique characteristics of equirectangular projection (ERP) panoramas, which map spherical data onto a 2D plane, violating the *i.i.d.* Gaussian noise assumption.

In addition to these fundamental issues, existing methods also contain redundant modeling, which introduces unnecessary computational costs. This redundancy manifests in several ways: 1) Some approaches [29, 36] iteratively inpaint and warp local areas, leading to error accumulation over successive steps. 2) Others [74, 55] adopt a dual-branch structure, with global and local branches, where the local branches are not directly used as final output targets. These challenges underscore the need for a straightforward, unified, and efficient paradigm for generating conditional panoramic images.

In this work, we explore a powerful yet less explored panoramic image generation paradigm to address the above limitations: the violation of the *i.i.d.* assumption and the lack of unification. Moreover, we allow the model to take text or images as input and control the generated content at any position on the canvas. As shown in Fig. 1, PAR is competent for text-to-panorama, panorama outpainting, and editing tasks. Notably, our PAR requires no task-specific data engineering - both conditions are seamlessly integrated via a single likelihood objective. In comparison, Omni[2] [68] relies on meticulously designed joint training datasets to align heterogeneous tasks, inevitably introducing complexity and domain bias.

For this goal, inspired by masked autoregressive modeling (MAR) [30], we propose PAR. On the one hand, it is based on AR modeling, which avoids the constraints of *i.i.d.* assumptions. On the other hand, it allows for generation in arbitrary order, overcoming the shortcoming of traditional raster-scan methods that are incapable of image-condition generation. This flexibility of PAR unifies conditional generation tasks - panorama outpainting reduces to a subset of text-to-panorama synthesis, where existing image tokens serve as partially observed sequences. Building upon the MAR-based generative framework, we introduce two critical enhancements to address inherent limitations in standard architectures. First, conventional VAE-based compression often exhibits spatial bias during encoding-decoding: center regions benefit from contextual information across all directions, while peripheral pixels suffer from unidirectional receptive fields, contradicting the cyclic nature of ERP panoramas. To mitigate this, we propose *dual-space circular padding*, applying cyclic padding operations in both latent and pixel spaces. This ensures seamless feature propagation across ERP boundaries, aligning geometric and semantic continuity. Second, as the foundation model is pre-trained on massive perspective images, we devise a *translation consistency loss*. This loss explicitly regularizes the model to accommodate the prior of ERP panoramas by minimizing discrepancies between the model's outputs under shifted and original inputs. Ultimately, two innovations holistically reconcile the unique demands of 360-degree imagery with the flexibility of MAR modeling. Experiments demonstrate that PAR outperforms specialist models in the text-to-panorama task, with 37.37 FID on the Matterport3D dataset. Moreover, on the outpainting task,

it has better generation quality and avoids the problem of repeated structure generation. Ablation studies demonstrate the model's scalability in terms of computational cost and parameters, as well as its generalization to out-of-distribution (OOD) data and other tasks, such as zero-shot image editing.

Our contribution can be summarized as follows: 1) We propose PAR, which fundamentally avoids the conflict between ERP and *i.i.d.* assumption raised by diffusion models, and also seamlessly integrates text and image-conditioned generation within a single architecture. 2) To enforce better adaptation to the panorama characteristic, we propose specialized designs, including dual-space circular padding and a translation consistency loss. 3) We evaluate our model on popular benchmarks, demonstrating competitive performance in text and image-conditional panoramic image generation tasks. The results also prove the scalability and generalization capabilities of our model.

## 2 Related Work

**Conditional Panoramic Image Generation.** There are two primary task paradigms in this area: 1) text-to-panorama (T2P), which synthesizes 360-degree scenes from textual descriptions, and 2) panorama outpainting (PO), which extends partial visual inputs to full panoramas. Given the success of diffusion models (DMs) [31, 32, 33, 43, 48, 51, 52] on generation tasks, such as image generation [48, 75, 23, 43, 42, 17], image editing [4, 37, 49], image super resolution [22, 50, 71], video generation [3, 20, 69], current approaches typically build upon DMs, yet face two problems. First, DM is inherently unsuitable for panorama tasks due to reliance on the *i.i.d.* Gaussian noise assumption, which ERP transformations violate. Second, existing approaches lack task unification. Specifically, T2P methods typically fine-tune SD, while PO solutions adapt the SD-inpainting variants. Although some work [68] attempts a unified solution, it is based on DM and relies on complex task-specific data engineering. From a methodological perspective, existing methods can be categorized into two main approaches: bottom-up and top-down. The former generates images with limited FoV sequentially [29, 36] or simultaneously [57, 27] and finally stitches them together, but faces the problem of lacking global information and error accumulation. The latter [74, 55] typically employs a dual-branch architecture comprising a panorama branch and a perspective branch. However, this design brings unnecessary computational cost. In parallel, several AR-based attempts have been made. They either rely on large DMs for final generation [36, 19], are limited to low resolution with additional up-sampling modules [10], or stitch perspective images without correct panorama geometry [80]. Overall, existing methods do not provide a unified and theoretically sound solution for conditional panorama generation, which we rethink through MAR modeling.

**Autoregressive Modeling.** Motivated by the success of large language models (LLMs) [5, 58], AR models, with great scalability and generalizability, are becoming a strong challenger to diffusion models. Several studies delve into visual AR with two distinct pipelines. One pipeline adopts raster-scan modeling [53, 47, 15, 16, 72, 63, 67, 28, 39], which treats the image as visual sentences and sequentially predicts the next token. However, this strategy does not fully exploit the spatial relationships of images. To make up for the shortcomings of raster-scan modeling, some works [7, 24, 73, 59, 30, 14, 1] extend the concept of masked generative models [8] to AR modeling, namely MAR. Instead of predicting one token within one forward, the model randomly selects several candidate positions and predicts attending to the decoded positions. Intuitively, this modeling provides flexible control over arbitrary spatial positions simply by adjusting the order of tokens to be decoded. Hence, we resort to MAR as a flexible way to unify text and image-conditioned panorama generation while avoiding the theoretical defects of panorama DMs.

## 3 Method

This section is organized into three parts. First, we introduce the preliminaries of panorama projection and review the standard conditional AR modeling. Second, we analyze the disadvantages of this approach for unified conditional generation. Last, we describe our PAR in the following sections, including vanilla MAR architecture, consistency alignment, and circular padding.

### 3.1 Preliminaries

**Equirectangular Projection** is a common parameterization for panoramic images. It establishes a bijection between spherical coordinates $(\theta, \phi)$ and 2D image coordinates $(u, v)$, where $\theta \in [0, \pi]$

denotes the polar angle (latitude) and $\phi \in (-\pi, \pi]$ represents the azimuthal angle (longitude). This projection preserves complete spherical visual information while maintaining compatibility with conventional 2D convolutional operations. Given a 3D point $\mathbf{P} = (x, y, z)$ on the unit sphere $\mathbb{S}^2$, we first compute its spherical coordinates and then obtain ERP coordinates $(u, v)$ via linear mapping:

$$\phi = \arctan\left(\frac{y}{x}\right), \quad \theta = \arccos(z), \quad u = \frac{\phi + \pi}{2\pi}W, \quad v = \frac{\theta}{\pi}H, \tag{1}$$

where $W \times H$ denotes the target image resolution.

**Conditional Autoregressive Models** formulate conditional data generation as a sequential process through next token prediction. Given an ordered sequence of tokens $\{x_t\}_{t=1}^T$, the joint distribution $p(x|c)$ factorizes into a product of conditional probabilities:

$$p(x|c) = \prod_{t=1}^T p(x_t|x_{<t}, c), \tag{2}$$

where $x_{<t} \equiv (x_1, ..., x_{t-1})$ denotes the historical context and $c$ indicates the conditions. This chain rule decomposition enables tractable maximum likelihood estimation, with the model parameters $\theta$ optimized to maximize:

$$\mathcal{L}(\theta) = \mathbb{E}_{x\sim\mathcal{D}}\left[\sum_{t=1}^T \log p_\theta(x_t|x_{<t}, c)\right], \tag{3}$$

where $\mathcal{D}$ represents the data distribution.

## 3.2 Rethinking Conditional Panoramic Image Generation

While DMs have demonstrated remarkable success in various image generation tasks, they rely on *i.i.d.* Gaussian noise assumption, which is violated by the spatial distortions introduced by the ERP transformation in Eq. 1, making diffusion-based methods inherently unsuitable for modeling panoramas. In this work, we therefore rethink conditional panorama generation from AR modeling, which is not constrained by such assumptions.

Building upon the standard conditional AR formulation in Eq. 2, we analyze text-to-panorama and panorama outpainting scenarios. The former takes text embeddings as $c$ while the latter takes observed image patches and optional textual prompts as conditions. In particular, this reformulates the two problems as follows:

$$p(x|c) = \prod_{t\in\mathcal{S}_u} p(x_t|x_{\mathcal{S}(t)}, c), \quad \mathcal{S}(t) \subseteq (\mathcal{S}_k \cup \{1, ..., t-1\}), \tag{4}$$

where $\mathcal{S}_k$ denotes known token positions from the condition and $\mathcal{S}_u = \{1, ..., T\} \setminus \mathcal{S}_k$ are targets. The context set $\mathcal{S}(t)$ dynamically aggregates:

$$\mathcal{S}(t) = \underbrace{\{j|j \in \mathcal{S}_k\}}_{\text{condition}} \cup \underbrace{\{j|j < t, j \in \mathcal{S}_u\}}_{\text{generated content}}, \tag{5}$$

This formulation generalizes text-to-panorama ($\mathcal{S}_k = \emptyset$) and outpainting ($\mathcal{S}_k \neq \emptyset$) scenarios. However, traditional raster-scan modeling is not competent for the panorama outpainting task, as $\mathcal{S}_k$ is not necessarily at the beginning of the sequence.

## 3.3 Masked Autoregressive Modeling with Consistency Alignment

We aim to learn a generalized AR model that naturally unifies text and image conditions and adapts to panoramic characteristics. An intuitive solution is MAR modeling, which allows generation in arbitrary order by marking the position of each token. However, existing foundation models are typically pre-trained on perspective images and cannot cope with panoramic scenarios. We delve into this problem and use the cyclic translation equivariance of ERP panoramas.

**Vanilla Architecture.** We start with a vanilla text-conditioned MAR model [14] to learn basic image generation without special constraints. Specifically, we use continuous tokens instead of discrete ones to reduce quantization error.

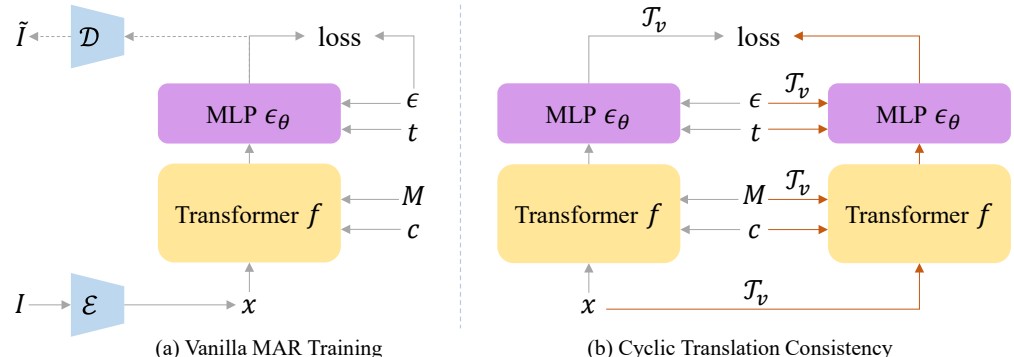

| (a) Vanilla MAR Training | (b) Cyclic Translation Consistency |

Figure 2: **Method illustration.** (a) PAR utilizes a transformer $f$ to predict regions obscured by mask $M$, then employs these predictions as conditions to drive an MLP $\epsilon_\theta$ in generating continuous tokens. The VAE encoder $\mathcal{E}$ and decoder $\mathcal{D}$ are frozen. The dashed line indicates the inference phase. $c$ and $t$ denote textual embeddings and time-steps, respectively. (b) Both original and $\mathcal{T}_v$-augmented triples $(x, \epsilon, M)$ are processed by the same model, and then aligned through a consistency loss.

As illustrated in Fig. 2 (a), panoramic images $I$ are first compressed into latent representations $x$ using a VAE encoder $\mathcal{E}$. These representations are fed into a transformer model $f$, which adopts an encoder-decoder architecture. Within the encoder, $x$ is divided into a sequence of visual tokens via patchification. A subset of these tokens is randomly masked by replacing the corresponding regions with a designated [MASK] token. Sine-cosine positional encodings (PEs) are added to the token sequence to preserve spatial information.

Simultaneously, textual prompts are processed by a pre-trained text encoder [26], which encodes them into textual embeddings $c \in \mathbb{R}^d$, where $d$ matches the dimensionality of the visual tokens. The unmasked visual tokens are fused with the textual embeddings $c$ in the encoder, and the resulting contextualized representations are then passed to the decoder, where they are integrated with the masked tokens to facilitate predictive reconstruction.

Unlike the standard AR modeling, the decoder's output is not used as the final result, but as a conditional signal $z$ to drive a lightweight denoising network, which is implemented by a multilayer perceptron (MLP) $\epsilon_\theta$. During the training phase, the noise-corrupted $x_t$ is denoised by $\epsilon_\theta$ under $z$ as follows:

$$\mathcal{L}_{va} = \mathbb{E}_{t,\epsilon} \left[ M \circ ||\epsilon_t - \epsilon_\theta(x_t|t, z)||^2 \right]. \tag{6}$$

In Eq. 6, $\circ$ is pixel-wise multiplication with binary mask $M$, where the value corresponding to the masked token is 1. The encoder-decoder $f$ and $\epsilon_\theta$ are jointly optimized.

**Cyclic Translation Consistency.** Current foundation models [48, 14] are predominantly trained on perspective images, leading to suboptimal feature representations for ERP-formatted panoramas. We introduce a cyclic translation operator $\mathcal{T}_v$ to address this domain gap and construct semantically equivalent panorama pairs, where $v$ indicates arbitrary translation distance. As shown in Fig. 2 (b), we construct an input pair, $(x, \epsilon, M)$ and $(x', \epsilon', M') = (\mathcal{T}_v(x), \mathcal{T}_v(\epsilon), \mathcal{T}_v(M))$. Both inputs share identical PE maps to prevent trivial solutions. The main model processes these pairs to produce outputs $y$ and $y'$, respectively. We enforce equivariance by aligning $\mathcal{T}_v(y)$ with $y'$ only on masked regions through the objective:

$$\mathcal{L}_{consistency} = M' \circ ||\mathcal{T}_v(\epsilon_\theta(x_t|t, x, M)) - \epsilon_\theta(x'_t|t, x', M')||^2, \tag{7}$$

The final learning objective can be written as follows:

$$\mathcal{L} = \mathcal{L}_{va} + \lambda \mathcal{L}_{consistency}, \tag{8}$$

where $\lambda$ is set as 0.1 in our experiments.

*Discussion.* The equivariance constraint holds exclusively for ERP panoramas due to their inherent horizontal continuity. For perspective images, applying $\mathcal{T}_v$ artificially creates discontinuous boundaries in the center of the shifted image, which violates two critical premises. 1) *Semantic equivalence.* Unlike panoramas, shifted perspective images lose semantic consistency due to non-periodic scene

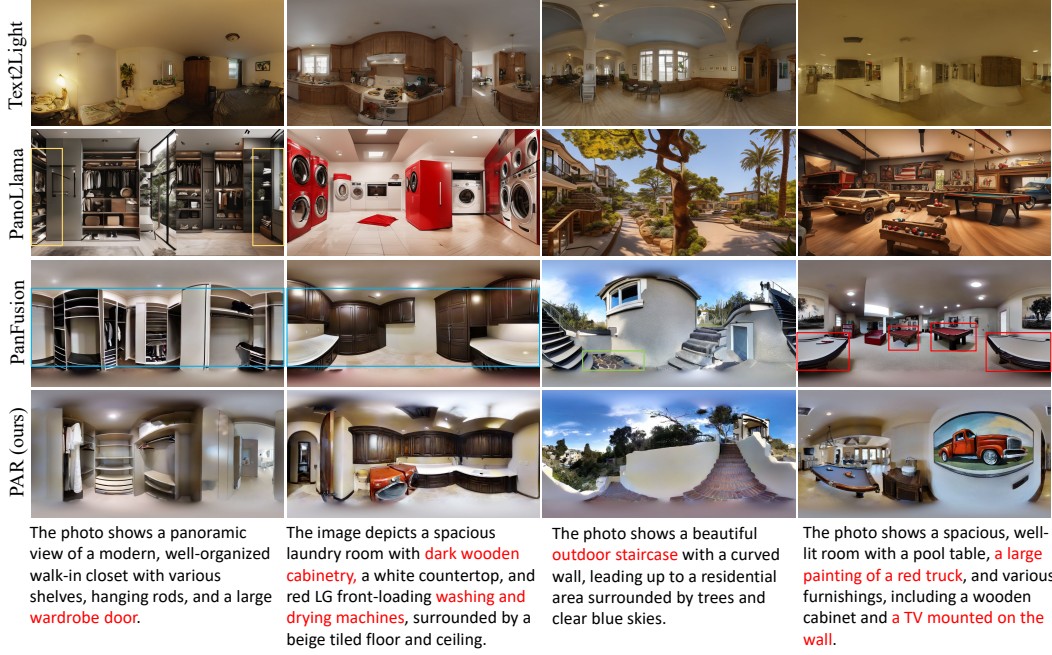

| | | | |
|---|---|---|---|
| The photo shows a panoramic view of a modern, well-organized walk-in closet with various shelves, hanging rods, and a large wardrobe door. | The image depicts a spacious laundry room with dark wooden cabinetry, a white countertop, and red LG front-loading washing and drying machines, surrounded by a beige tiled floor and ceiling. | The photo shows a beautiful outdoor staircase with a curved wall, leading up to a residential area surrounded by trees and clear blue skies. | The photo shows a spacious, well-lit room with a pool table, a large painting of a red truck, and various furnishings, including a wooden cabinet and a TV mounted on the wall. |

Figure 3: **Visual comparisons with previous methods on text-to-panorama task.** Previous methods neglect the circular consistency characteristics (yellow box), or suffer from repetitive objects (red box), artifact generation (blue box), and inconsistent combinations (green box). The highlighted portions in captions indicate text-image alignment failure cases of baseline methods.

structures. 2) *Model prior*. Generation models inherently enforce feature continuity in image interiors, making it infeasible to produce fragmented content.

**Circular Padding.** While the AR model incorporates panoramic priors, the VAE's latent space processing may still introduce discontinuities. Standard convolution operations in VAE provide sufficient context for central regions but suffer from information asymmetry at image edges - pixels near boundaries receive unidirectional receptive fields. To address this issue, we propose a dual-space circular padding scheme: pre-padding and post-padding. The former is applied in pixel space for the VAE encoder, while the latter is used in latent space for the VAE decoder. Specifically, we crop a particular portion of the left and right regions and concatenate them to the other side, as formulated in Eq. 9.

$$C_r(x) = \text{concat}(x[..., -rW/2 :], x, x[..., : rW/2]), \tag{9}$$

where $x \in \mathbb{R}^{H \times W}$ and $C_r(x) \in \mathbb{R}^{H \times (1+r)W}$, indicate the original and the padded tensor, respectively. $r$ indicates the ratio of padding width to that of the original images or latent variables. After the VAE transformation, the edge positions have sufficient context information. Therefore, the areas that belong to the padding are discarded.

# 4 Experiment

## 4.1 Experiment Settings

**Implementation Details.** We utilize NOVA [14] as the initialization and set the resolution as $512 \times 1024$. Our model is trained for 20K iterations with a batch size 32. We employ an AdamW optimizer [35]. The learning rate is $5 \times 10^{-5}$ with the linear scheduler. In the inference stage, we set the CFG [23] coefficient as 5. In this paper, the masking sequence for training and the sampling sequence for inference are initialized with a uniform distribution unless otherwise stated.

**Datasets and Metrics.** We mainly use Matterport3D [6] for comparisons. The split of the training and validation set follows PanFusion [74]. We use Janus-Pro-7B [9] to generate the captions.

Table 1: Performance on text-to-panorama task. PO and PE represent panorama outpainting and editing, respectively. **Bold** and underline indicate the first and second best entries.

| Modeling | Method | #params | T2P | PO | PE | FAED ↓ | FID ↓ | CLIP Score ↑ | DS ↓ |
|---|---|---|---|---|---|---|---|---|---|
| Diffusion | PanFusion [74] | 1.9B | ✓ | - | - | 5.12 | 45.21 | 31.07 | 0.71 |
| AR | Text2Light [10] | 0.8B | ✓ | - | - | 68.90 | 70.42 | 27.90 | 7.55 |
| | PanoLlama [80] | 0.8B | ✓ | - | - | 33.15 | 103.51 | **32.54** | 13.99 |
| | PAR (ours) | 0.3B | ✓ | ✓ | ✓ | 3.39 | 41.15 | 30.21 | 0.58 |
| | PAR (ours) | 0.6B | ✓ | ✓ | ✓ | **3.34** | 39.31 | 30.34 | **0.57** |
| | PAR (ours) | 1.4B | ✓ | ✓ | ✓ | 3.75 | **37.37** | 30.41 | 0.58 |

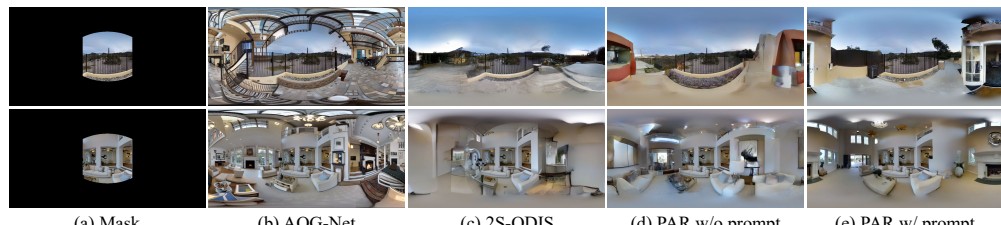

| (a) Mask | (b) AOG-Net | (c) 2S-ODIS | (d) PAR w/o prompt | (e) PAR w/ prompt |
|---|---|---|---|---|

Figure 4: **Qualitative comparisons of panorama outpainting on the Matterport3D dataset.** PAR-1.4B is used for this task, where PAR w/o prompt means the textual prompt is set as empty.

Several metrics are used in this paper. Fréchet Inception Distance (FID) [21] measures the similarity between the distribution of real and generated images in a feature space derived from a pre-trained inception network. Lower FID scores indicate better realism of synthesized results. CLIP Score (CS) [46] evaluates the alignment between text and image. Furthermore, as FID relies on an Inception network [54] trained on perspective images, we also report the Fréchet Auto-Encoder Distance (FAED) [41] score, which is a variant of FID customized for panorama. To measure the cycle consistency of panoramic images, we adopt Discontinuity Score (DS) [11].

**Baselines.** Since only a few approaches combine T2P and PO, we compare PAR with specialist methods separately. Specifically, for T2P, we compare with several diffusion-based and AR-based methods, including PanFusion [74], Text2Light [10], and PanoLlama [80]. We take AOG-Net [36] and 2S-ODIS [38] as PO baselines. All experiments are implemented on the Matterport3D dataset.

## 4.2 Main Results

**Text-to-Panorama.** Tab. 1 shows the quantitative comparison results. Within the scope of AR-based methods, our PAR significantly outperforms previous approaches regarding generation quality and continuity score. Our method is slightly inferior in the CLIP score metric, probably because CLIP [46] is pre-trained on massive perspective images, and panoramic geometry is not well aligned with perspective images. Moreover, our model performs decently compared with a strong diffusion-based baseline, PanFusion, with only 0.3B parameters. Specifically, PAR-0.3B outperforms PanFusion with 1.73 and 4.06 on FAED and FID, with a discontinuity score of 0.58. Scaling up the parameters helps achieve better results, which is further illustrated in Sec. 4.3.

Fig. 3 provides visual comparisons with the aforementioned baseline methods, including diffusion-based and AR-based models. Our model shows better instruction-following ability, more consistent generation results, and avoids duplicate generation. Specifically, Text2Light cannot obtain satisfactory synthesized results. PanoLlama cannot understand panorama geometry and neglects the characteristic of cycle consistency. PanFusion produces more reasonable results, but sometimes faces artifact problems. For example, the first and second columns of the third row show a room without an exit.

**Panorama Outpainting.** Tab. 2 compares our method with AOG-Net and 2S-ODIS. We use the model trained with Eq. 8 and inference following Eq. 4, where $\mathcal{S}_k$ indicates the positions within the mask. Our model achieves 32.68 and 12.20 on FID and FID-h, respectively. FID-h is similar to FID, but the difference is that it extracts eight perspective views in the horizontal direction with a 90-degree FoV for each panoramic image, which pays more attention to the generation quality of local regions. Fig. 4 visualizes the synthesized results. Existing methods suffer from local distortion

Table 2: **Panorama outpainting results on the Matterport3D dataset.** For FID-h, we horizontally sample 8 perspective images with FoV=90° for each panorama and then calculate their FID.

| Method | T2P | PO | PE | FID ↓ | FID-h ↓ |
|---|---|---|---|---|---|
| AOG-Net [36] | - | ✓ | - | 83.02 | 37.88 |
| 2S-ODIS [38] | - | ✓ | - | 52.59 | 35.18 |
| PAR w/o prompt | ✓ | ✓ | ✓ | 41.63 | 25.97 |
| PAR w/ prompt | ✓ | ✓ | ✓ | **32.68** | **12.20** |

Scaling up training compute →

Scaling up model parameters

Figure 5: **Scaling parameters and training compute improve fidelity and soundness**. Two cases are shown with 3 model sizes and 3 different training stages. From top to bottom: 0.3B, 0.6B, 1.4B. From left to right: 25%, 50%, 100% of the training process.

(the $3^{rd}$ column) or counterfactual repetitive structures (the $2^{nd}$ column). Our method overcomes these problems and still produces reasonable results without textual prompts.

## 4.3 Ablation Study

**Scalability.** We train with three different parameter sizes, namely 0.3B, 0.6B, and 1.4B, respectively. In Tab. 1, FID and CS improve with increasing parameters. We also visualize the performance under different training stages, including 25%, 50%, and 100% of the whole training process. As shown in Fig. 5, the improvement of fidelity and soundness is consistent with the scaling of parameters and computation. Larger models learn data distributions better, such as details and panoramic geometry.

**Generalization.** Fig. 6 visualizes the outpainting results of PAR on OOD data. SUN360 [66] is used for evaluation, which has different data distributions from Matterport3D. PAR shows decent performance across various scenarios. We also compared with several PO baselines trained on the SUN360 dataset. However, Diffusion360 [18] suffers from unrealistic scenes and lacks details. Panodiff [62] also encounters artifact problems (red sky in the $1^{st}$ row).

PAR can also adapt to different tasks, such as panoramic image editing. As shown in Fig. 7, the model needs to preserve the content inside the mask and regenerate according to new textual prompts. We designed three challenging text prompts, whose content differs from the training datasets, and contains specific conflicts with the original image. For example, fire *vs.* yard in (b), snow *vs.* current season of scene in (c), and ocean *vs.* current place of the scene in (d). Interestingly, our model can generate them well and understands panoramic characteristics. As shown in Fig. 7 (b), the left and right sides of the image can be stitched to form a whole fire.

**Translation Equivariance.** Tab. 3 demonstrates the effectiveness of consistency loss. Eq. 7 helps the model adapt to panoramic characteristics, thus improving generation quality. Fig. 8 (left) shows the convergence of $r$. Post-padding significantly improves DS when $r_{post} > 0$ as it helps to smooth stitching seams in the pixel space. However, the model without pre-padding suffers from higher discontinuity, indicating the misalignment between the training and testing phases. With

Table 3: Ablations on cyclic consistency loss.

| Method | FID ↓ | CLIP Score ↑ | DS ↓ |
|---|---|---|---|
| w/o $\mathcal{L}_{consistency}$ | 39.55 | 30.25 | 0.57 |
| w/ $\mathcal{L}_{consistency}$ | 37.37 | 30.41 | 0.58 |

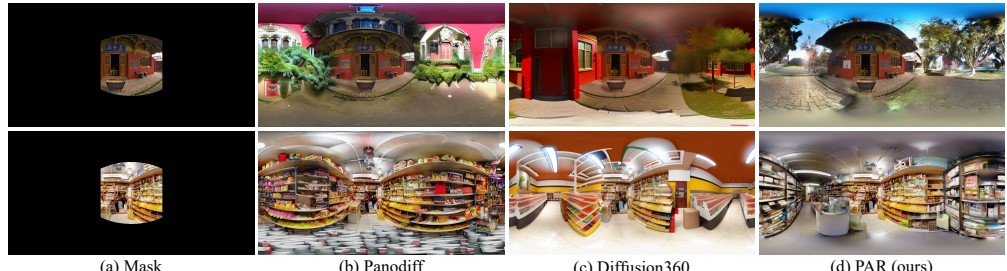

| (a) Mask | (b) Panodiff | (c) Diffusion360 | (d) PAR (ours) |

Figure 6: **Panorama outpainting on OOD dataset.** The images are from SUN360, which is out of the distribution of our training data. PAR generates realistic panorama images while previous methods have problems like artifacts, or unrealistic results.

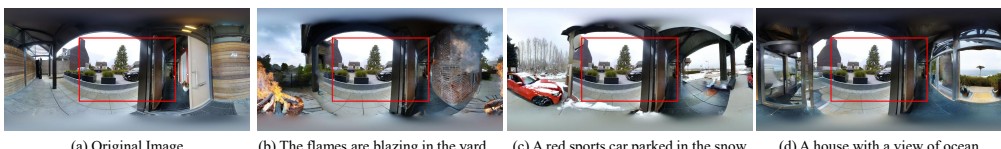

| (a) Original Image | (b) The flames are blazing in the yard | (c) A red sports car parked in the snow | (d) A house with a view of ocean |

Figure 7: **Panoramic image editing**. We design several challenging textual prompts to enforce the model to regenerate the contents outside the red box while maintaining the inside contents. Note that our PAR performs zero-shot task generalization.

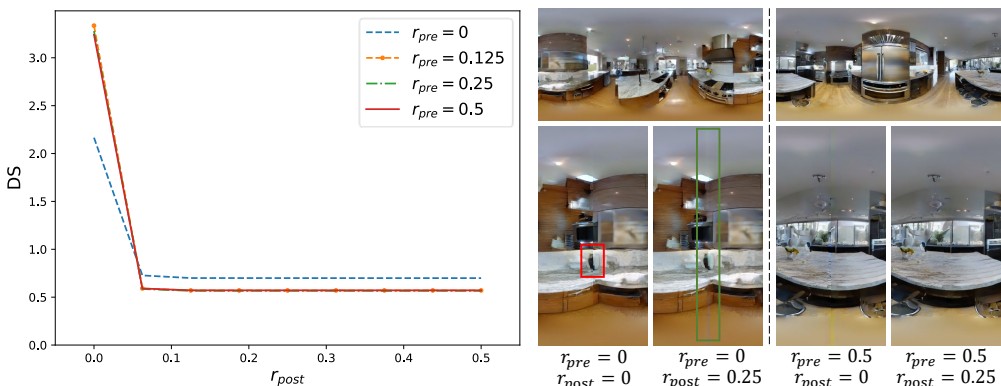

Figure 8: **Left**: The curve of DS about $r_{post}$. **Right**: Visualizations of different circular padding strategies. Each part consists of three images, where the top one represents the panoramic image generation result, the left and right ones represent the edge stitching results, respectively. The red box indicates the semantic discontinuity, and the green box highlights the seam after post-padding.

the same $r_{pre}$, DS converges quickly when $r_{post} > 0.125$. A similar phenomenon is observed with $r_{post}$ fixed. Specifically, the curve of $r_{pre} = 0.25$ and $r_{pre} = 0.5$ almost coincide.

Fig. 8 (right) visually explains the difference between the two padding strategies. When $r_{pre} = 0$, semantic inconsistency exists between the left and right edges of the panoramic image. Post-padding cannot repair it even with a large padding ratio, and the green box bounds the smoothed seam. In contrast, the model generates consistent contents when pre-padding is applied.

## 5 Conclusion

This paper proposes PAR, a MAR-based architecture that unifies text- and image-conditioned panoramic image generation. We delve into the cyclic translation equivariance of ERP panoramas and correspondingly propose a consistency alignment and circular padding strategy. Experiments demonstrate PAR's effectiveness on text-to-panorama and panorama outpainting tasks, whilst showing

promising scalability and generalization capability. We hope our research provides a new solution for the panorama generation community.

**Acknowledgement.** This work was supported by the National Key Research and Development Program of China (No. 2023YFC3807600).

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

# A   Proof of Non-*i.i.d.* of ERP Transformation

## A.1   Problem Setting

Let the unit sphere

$$\mathbb{S}^2 = \{(x, y, z) \in \mathbb{R}^3 \ : \ x^2 + y^2 + z^2 = 1\}$$

be parameterised by spherical angles $(\phi, \theta) \in (-\pi, \pi] \times [0, \pi]$. The equirectangular projection of spatial resolution $(H, W) \in \mathbb{N}^2$ is the mapping

$$T : [0, W) \times [0, H) \longrightarrow (-\pi, \pi] \times [0, \pi], \qquad (u, v) \mapsto \Big(\phi(u), \theta(v)\Big)$$

with

$$\phi(u) = \frac{2\pi u}{W} - \pi, \qquad \theta(v) = \frac{\pi v}{H}.$$

Conversely, $u = \dfrac{\phi + \pi}{2\pi} W, \ v = \dfrac{\theta}{\pi} H.$

## A.2   Gaussian noise field on the sphere

Let $\big\{\, \varepsilon(\phi, \theta)\, \big\}_{(\phi, \theta) \in (-\pi, \pi] \times [0, \pi]}$ be a Gaussian noise field on the sphere, i.e.

$$\mathbb{E}[\varepsilon(\phi, \theta)] = 0, \qquad \mathrm{Cov}\big(\varepsilon(\phi, \theta), \varepsilon(\phi', \theta')\big) = \sigma^2\, \delta(\phi - \phi')\, \delta(\theta - \theta'), \tag{10}$$

where $\delta$ is the Dirac distribution and $\sigma^2 > 0$ is a constant variance density (per steradian).

## A.3   Pixel value definition in ERP

Each ERP pixel $(u, v) \in \{0, \dots, W-1\} \times \{0, \dots, H-1\}$ covers a spherical patch

$$P_{uv} \ = \ \Big[\phi(u - \tfrac{1}{2}), \phi(u + \tfrac{1}{2})\Big) \times \Big[\theta(v - \tfrac{1}{2}), \theta(v + \tfrac{1}{2})\Big).$$

Its area is

$$A_{uv} = \int_{P_{uv}} \sin\theta\, d\phi\, d\theta \ = \ \frac{2\pi}{W}\Big[\cos\theta(v - \tfrac{1}{2}) - \cos\theta(v + \tfrac{1}{2})\Big] \ = \ \frac{2\pi^2}{WH}\, \sin\theta_v, \tag{11}$$

with the mid-latitude $\theta_v := \theta(v) = \pi v / H$.

The *noise value stored in the pixel* is the area–average

$$\eta_{uv} := \frac{1}{A_{uv}} \int_{P_{uv}} \varepsilon(\phi, \theta)\, dA, \quad dA = \sin\theta\, d\phi\, d\theta. \tag{12}$$

Because the integral in Eq. 12 is a *linear* functional of the Gaussian field $\varepsilon$, each $\eta_{uv}$ is still Gaussian.

## A.4   Independence

For two different pixels $(u, v) \neq (u', v')$ the patches $P_{uv}, P_{u'v'}$ are disjoint. Since Eq. 10 states that $\varepsilon(\phi, \theta)$ is white, integrals over disjoint regions are uncorrelated, hence Gaussian independent:

$$\mathrm{Cov}(\eta_{uv}, \eta_{u'v'}) = \iint_{P_{uv}} \iint_{P_{u'v'}} \frac{\sigma^2\, \delta(\phi - \phi')\delta(\theta - \theta')}{A_{uv} A_{u'v'}}\, dA\, dA' = 0 \quad \Longrightarrow \quad \eta_{uv} \perp\!\!\!\perp \eta_{u'v'}. \tag{13}$$

## A.5 Non-identical distribution

The variance of one pixel, using Eq. 10, Eq. 11, and Eq. 12, is

$$\text{Var}(\eta_{uv}) = \frac{1}{A_{uv}^2} \iint_{P_{uv}} \iint_{P_{uv}} \sigma^2 \, \delta(\phi - \phi')\delta(\theta - \theta') \, dA \, dA' = \frac{\sigma^2}{A_{uv}} = \sigma^2 \, \frac{WH}{2\pi^2} \, \frac{1}{\sin\theta_v}. \quad (14)$$

Eq. 14 depends on the latitude index $v$ through the factor $\sin\theta_v$. Pixels close to the poles ($\theta_v \approx 0$ or $\pi$) possess a larger variance than pixels near the equator ($\theta_v \approx \pi/2$). Hence

$$\text{Var}(\eta_{uv}) \neq \text{Var}(\eta_{u,v'}) \quad \text{for } v \neq v'.$$

## B More Details on Method

**Implementation Details.** The sampling step for PAR is set to 64 and $r = 0.125$ by default. We use Janus Pro-7B to get the captions with the template as "Use one sentence to describe the photo in detail.". We then calculate the token lengths and keep them less than 77. Experiments shows that less than $0.1\%$ of the captions do not meet the requirements, and we add "and briefly" at the end of those prompts.

**Framework Details.** The transformer takes patchified visual tokens (from the VAE encoder), masking indicators, and text embeddings as inputs. It outputs a conditional signal to drive the subsequent denoising network MLP. The decoding mechanism is illustrated in Fig. 9.

The MLP has multiple stacked blocks. Within each block, the input is first processed by AdaLNZero to get gating coefficients. It is then transformed using a projection layer consisting of two linear layers with silu activation. The result is then LN-normalized and multiplied by the gating coefficients, and finally combined with a residual connection.

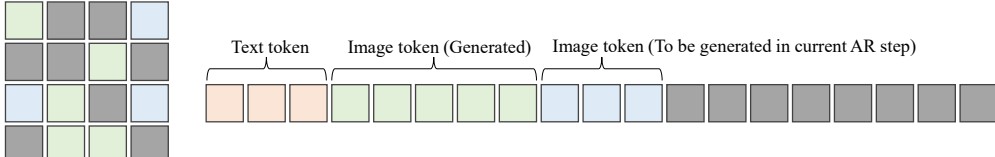

Figure 9: Illustration of decoding mechanism.

## C More Empirical Studies and Visual Results

**Time Analysis.** It takes about 2 days to train the 1.4B model on 8 NVIDIA A100 GPUs. As shown in Tab. 4, we benchmark the inference speed for the 0.3B model with different AR steps using one NVIDIA A100 GPU with a batch size of 8. The denoising steps for MLP are 25.

Table 4: Ablations of inference time under different AR steps.

| AR steps | 16 | 32 | 64 |
|---|---|---|---|
| Inference speed (sec/image) ↓ | 3.02 | 5.49 | 10.03 |

We also compare PAR-0.3B with PanFusion with 64 AR steps in Tab. 5.

Table 5: Comparison with PanFusion regarding inference time.

| | PanFusion | PAR (ours) |
|---|---|---|
| Inference speed (sec/image) ↓ | 28.91 | 10.03 |
| FID ↓ | 45.21 | 41.15 |

Table 6: Ablations of different padding ratios regarding inference time.

| Padding Ratio | 0 | 0.0625 | 0.125 | 0.25 | 0.5 |
|---|---|---|---|---|---|
| Inference speed (sec/image) ↓ | 2.99 | 3.01 | 3.02 | 3.04 | 3.05 |
| Additional costs | 0 | 0.67% | 1.00% | 1.67% | 2.01% |

Table 7: Text-to-panorama results on the Matterport3D dataset.

| | FAED ↓ | FID ↓ | CLIP Score ↑ | DS ↓ |
|---|---|---|---|---|
| DiffPano [70] | 10.03 | 53.29 | 30.31 | 6.16 |
| UniPano [40] | 5.87 | 44.74 | 30.45 | 0.77 |
| PAR-0.3B (ours) | 3.39 | 41.15 | 30.21 | 0.58 |

As shown in Tab. 6, circular padding does not bring about additional computational cost in the transformer and MLP, which account for the majority of inference time. The AR step is 16.

**More Results on Text-to-Panorama.** Tab. 7 shows that our method has advantages on text-to-panorama task. We also provide more visual results in Fig. 10. Our model can generate reasonable panoramic images under the guidance of textual prompts. Tab. 9 provides ablation study for different denoising steps with our 0.3B model.

**More Results on Panorama Outpainting.** Tab. 8 shows that our method has advantages on panorama outpainting task. We provide more visual results for panorama outpainting in Fig. 11. We test the performance of our model in indoor setting and outdoor setting, and our model can achieve decent results, which proves its generalizability.

**Visual Results of Circular Padding.** Fig. 12 compares different circular padding strategies, including $r_{pre} = 0$ and $r_{pre} = 0.5$. $r_{post}$ ranges from 0 to 0.5. It converges quickly with $r_{post} > 0$. $r_{pre}$ helps to maintain cycle consistency at the semantic level.

**Classifier-free Guidance.** We evaluate PAR-1.4B with different CFG coefficients. The model obtains 40.04 FID when CFG=3 and 39.76 FID when CFG = 10, indicating that too large or too small guidance strength deteriorates the quality of the generation.

**Ablations on other datasets.** We include additional experiments using the Structured3D [78] dataset, which is a large-scale synthesized indoor dataset for house design with well-preserved poles. We sampled 9000 images for training and 1000 for testing. In Tab. 10, our method outperforms PanoLlama. Fig. 14 visualizes the synthesized results of PAR-0.3B. This indicates that the blur in the pole regions originates from the dataset and is irrelevant to the model, which can also be demonstrated by the visualizations of perspective projections in Fig. 13.

**Zero-shot inference on OOD datasets.** We evaluate panorama discontinuity using both our method and StitchDiffusion [61]. StitchDiffusion forces the model to fuse seam areas during inference. On the contrary, our pre-padding makes the model learn semantic consistency during the training phase, and post-padding achieves pixel-level consistency in the inference stage. In this experiment, PAR-0.3B and StitchDiffusion generate images for 100 text prompts sampled from a subset of SUN360, respectively. Our method achieves a DS score of 0.63, outperforming StitchDiffusion's 1.12.

For zero-shot outpainting, we compare the two methods on 100 images from the Structured3D dataset, which is excluded from both training sets for fairness. As shown in Tab. 11, our model outperforms Diffusion360.

**Failure case.** As shown in Fig. 15, our model may fail in some details of small objects, such as tables and sofas.

# D  Discussion

**Comparisons to multi-view diffusion models.** On the one hand, our AR-based method has several advantages over multi-view diffusion models (MVDMs). First, Computational Efficiency. MVDMs require generating several perspective images to cover a panorama, resulting in more computational

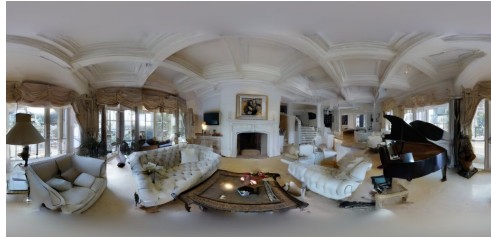
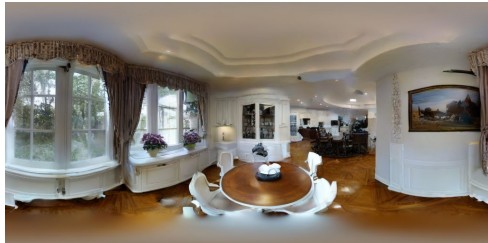

The image depicts a spacious, elegantly designed living room with large windows, white walls, and a mix of modern and traditional furniture, including sofas, a coffee table, and a grand piano.

The image depicts a spacious, elegant kitchen and dining area with a round wooden table, white chairs, and a large window adorned with curtains, showcasing a blend of modern and classic design elements.

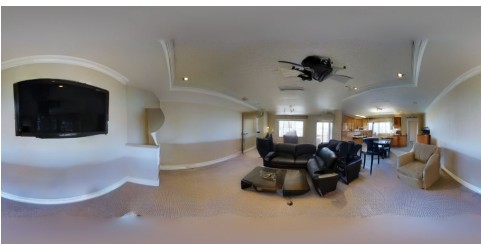
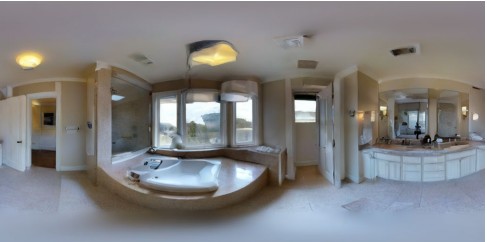

The image depicts a spacious, well-lit living room with a mounted TV, comfortable seating, and a view of the kitchen and dining area in the background.

The image depicts a spacious, modern bathroom with a large bathtub, double vanity, and large windows allowing natural light to fill the room.

Figure 10: Visualization of text-to-panorama generation, including the generated results and corresponding textual prompts.

Table 8: Panorama outpainting results on the Matterport3D dataset.

|  | FID ↓ | FID-h ↓ |
| --- | --- | --- |
| PanoDiff [62] | 65.11 | 46.58 |
| PAR w/o prompt | 41.63 | 25.97 |

cost due to the attention mechanism. Second, Global Consistency. Coordinating multiple views demands complex attention mechanisms to ensure coherence, and struggles to maintain a consistent global understanding - sometimes leading to duplicated or inconsistent content. Third, Fewer Seam Issues. MVDMs must handle more seams at the boundaries between views, which increases the difficulty of achieving seamless stitching and can introduce visible artifacts.

On the other hand, since our model directly generates the entire panoramic image, it needs to use panoramic data to drive the pre-trained model (trained on perspective images) to adapt to the panoramic domain knowledge, which might be less efficient.

**Broader Impacts.** Our proposed PAR employs autoregressive set-by-set modeling for panoramic image generation tasks. It has significant advantages in generation performance compared with previous methods. Moreover, it shows the capability of zero-shot task generalization, like panorama outpainting and editing. This is an attempt towards the unification model in the field of panoramic image generation and will motivate people to explore the unification of different tasks in panorama generation. From the perspective of social impact, this will help the creation of artistic content, but at the same time it may result in the problem of fake content.

**Safeguards.** The content generated by the generative model, including AR models in this paper, demonstrates a high degree of correlation with the training data. From this perspective, it is important to ensure the fairness and cleanliness of the training data. In this way, we can effectively prevent the model from generating harmful content.

**Limitations.** We find that there is still a gap between the generated results and the real panoramic images, such as some details and textures. We argue that scaling on more realistic images can help alleviate this problem. However, due to the scarcity of panoramic data, we leave it as future work.

Table 9: Ablations of denoising steps.

| Steps | 10 | 20 | 30 | 40 |
|---|---|---|---|---|
| FID ↓ | 41.57 | 40.19 | 41.98 | 43.77 |

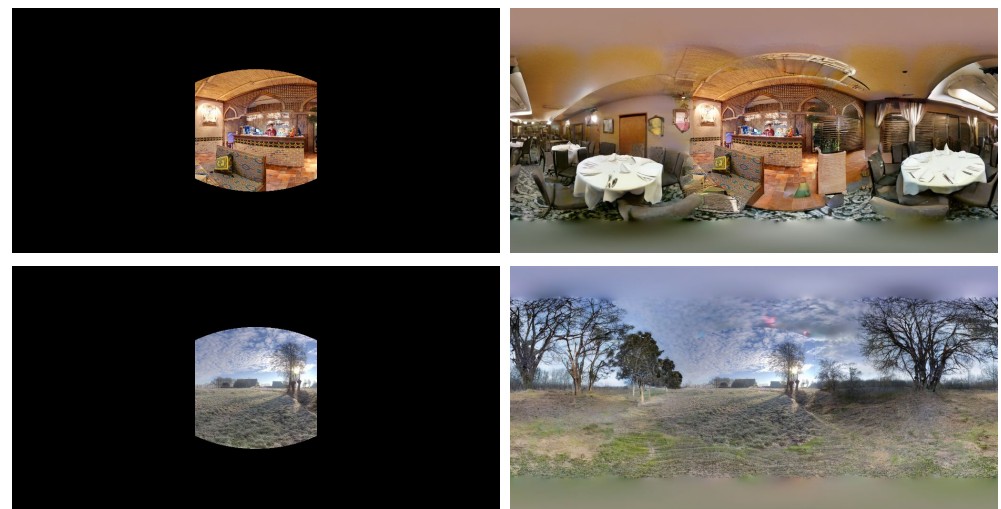

Figure 11: Visualization of panorama outpainting results.

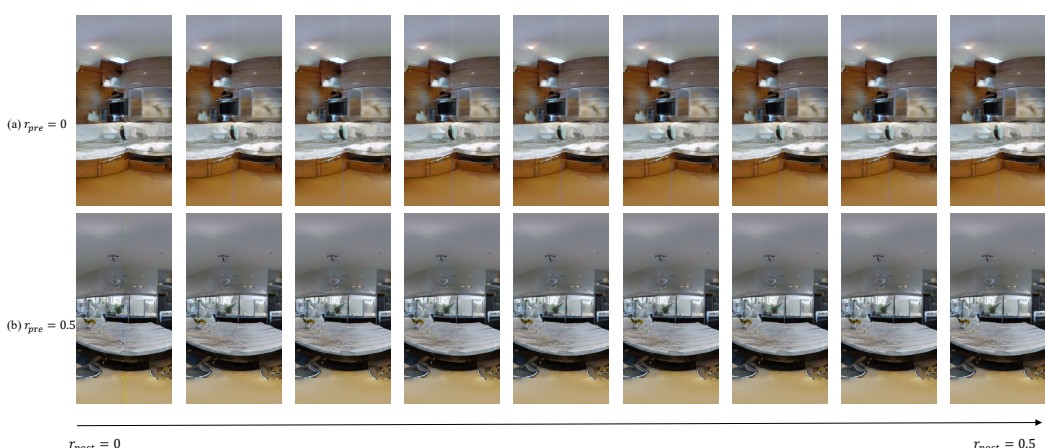

Figure 12: Visualization results of circular padding under different settings.

Table 10: Comparisons of PAR-0.3B and PanoLlama on Structured3D dataset.

| | FID ↓ | CLIP Score ↑ | DS ↓ |
|---|---|---|---|
| PanoLlama [80] | 125.35 | 33.90 | 16.37 |
| PAR-0.3B | 47.02 | 30.75 | 0.68 |

Table 11: Zero-shot panorama outpainting results.

| | FID ↓ | FID-h ↓ |
|---|---|---|
| Diffusion360 [18] | 140.91 | 74.89 |
| PAR w/o prompt | 127.01 | 68.27 |

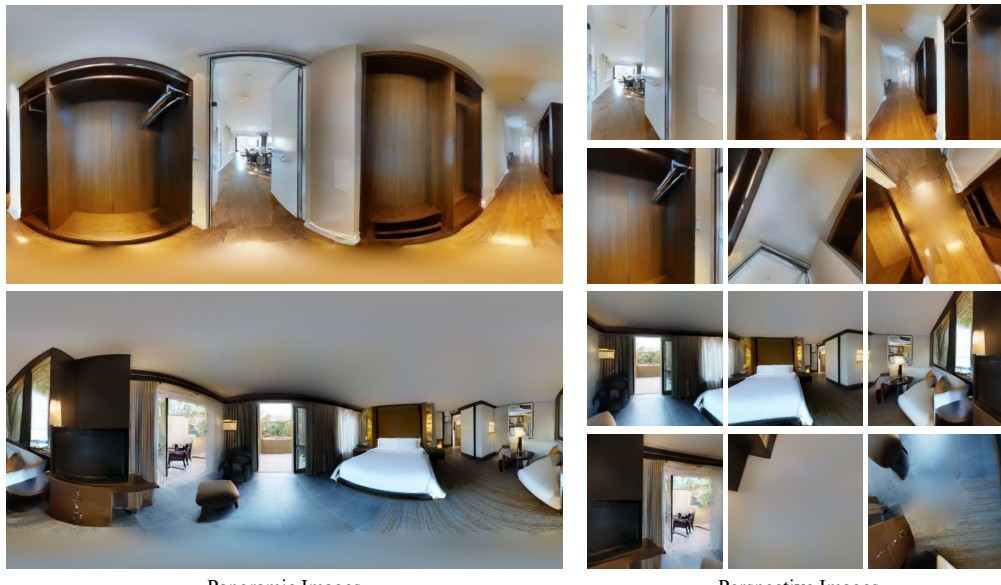

Panoramic Images                              Perspective Images

Figure 13: Perspective projections of synthesized results on Matterport3D dataset.

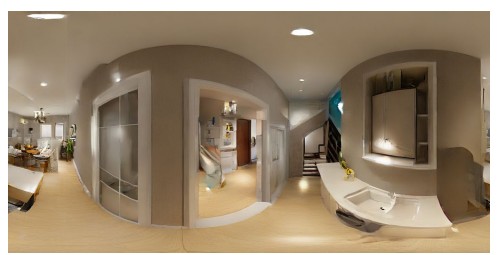

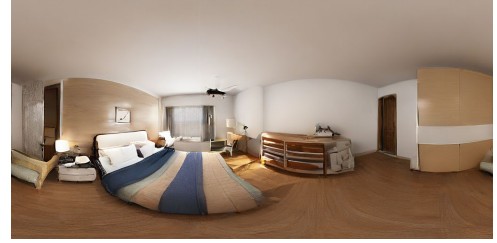

The image depicts a modern, well-lit interior of a home with a kitchen, dining area, hallway, living room, and staircase, showcasing a warm and inviting atmosphere.

The image depicts a modern, well-organized bedroom with a large bed, wooden flooring, and a variety of furniture including a wardrobe, a chair, a desk, and a dresser.

Figure 14: Visualizations of synthesized results on Structured3D dataset.

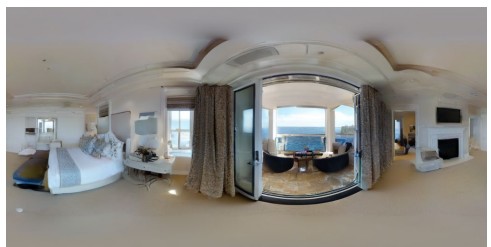

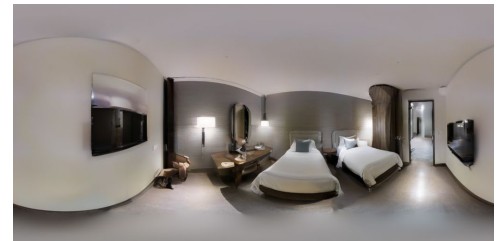

(a) The photo depicts a spacious, modern bedroom with a large bed, a sitting area, and a balcony overlooking a scenic view of the ocean.

(b) The image depicts a modern, spacious hotel room with two twin beds, sleek furniture, and a clean, neutral color palette.

Figure 15: Failure cases.

