# OpenReview forum: "Conditional Panoramic Image Generation via Masked Autoregressive Modeling"
_NeurIPS.cc/2025/Conference — NeurIPS 2025 poster_

### Official Review · Reviewer_1sbD · 2025-07-01

**Clarity:** 2
**Significance:** 2
**Originality:** 2
**Rating:** 4
**Confidence:** 5

**Summary:**

This paper introduces the Panoramic AutoRegressive (PAR) model, a panoramic image generation framework based on masked autoregressive modeling. Previous methods based on diffusion models had critical limitations that a single model couldn’t handle both panorama generation and outpainting tasks, and that they faced the constraints of iid assumptions which is inherently violated by the equirectangular projection during panorama generation. To resolve these issues, the authors utilize a conditional masked autoregressive modeling framework that removes the iid assumptions and also handles generation and outpainting tasks in an unified manner. Moreover, PAR suggests novel techniques tailored for panorama generation, including dual-space circular padding and a translation consistency loss. Consequently, PAR achieves competitive results compared to previous state-of-the-art diffusion-based methods and shows strong scalability and zero-shot generalization capabilities.

**Questions:**

* Although the paper is well-structured, the overuse of abbreviations seems to harm its readability. For example, using "T2P" for "text-to-panorama", "CS" for "CLIP score", "DS" for "discontinuity score" were often confusing while reading the paper and had to revisit the original definition repeatedly.

* Figure 2 doesn't seem very informative for understanding the architecture of PAR. Providing a more detailed visual overview of the proposed framework would help the authors understand the main idea. For instance, it would be better to clarify how the input and output of the "Transformer f" are structured, instead of solely representing it using a letter.

**Ethical Concerns:**

["NO or VERY MINOR ethics concerns only"]

**Final Justification:**

**(After Initial Rebuttal)**

While the authors' rebuttal has provided clarifications on most of the raised concerns, multiple concerns regarding the methods novelty and the comparison with previous methods yet remain unresolved.

* **[Q1]** Missing the citation of StitchDiffusion [1] and not discussing the difference/similarity with their padding technique in the original manuscript remains a critical limitation and raises questions about the novelty and completeness of the proposed method. Moreover, the third point made by the authors is not convincing, since the naive comparison with different settings in different works cannot convey a fair comparison. Without a systematic evalution or ablation, it is difficult to support the authors' claims.

* **[Q2]** The additional experiments provided are not helpful in understanding the advantage of PAR, because (1) the evaluation only contains PanoDiff while the authors discussed both PanoDiff and Diffusion360, and (2) the evaluation metrics used in the manuscript are missing, and only FID is used.

**(After Additional Discussions)**

The additional clarifications and experiment results resolved my remaining concerns above. Therefore I raise the rating to **borderline accept (4)**.

---

[1] Customizing 360-Degree Panoramas through Text-to-Image Diffusion Models, WACV 2024.

**Limitations:**

Yes

**Paper Formatting Concerns:**

No concerns

**Quality:**

3

**Strengths And Weaknesses:**

**Strength**

* The paper is clearly written and easy to follow.

* Adopting masked autoregressive modeling for panoramic generation is a reasonable design choice given the limitations of previous diffusion-based approaches.

* PAR outperforms the main baselines in both panorama generation and outpainting tasks, and the qualitative comparisons further show its advantage over previous methods.

* Thorough ablation studies for the proposed techniques - circular padding and consistency loss - make it clear that the design choices in PAR lead to the intended outputs.

**Weakness**

* The evaluation setup for panorama outpainting is confusing. Why were AOG-Net and 2S-ODIS used as baselines for the quantitative comparion, yet PanoDiff and Diffusion360 were compared on the OOD dataset? Was it unavailable to conduct quantitative comparisons with PanoDiff and Diffusion360 on the Matterport3D dataset?

* I couldn't find any visualizations of **perspective projections** of the generated panoramas. I believe some visualizations along with qualitative comparisons on the perspective projections are necessary to versity whether the generated outputs from PAR respect the geometric properties of equirectangular panoramas.

* Some previous works on panoramic generation are not compared as baseline, while they are included in the related work. Both DiffPano [1] and CubeDiff [2] were mentioned in the related work section for **Conditional Panoramic Image Generation**. I am curious if the authors could not compare with these methods due to the unavailability of their code and models.

* The idea of adding paddings to the left and right sides to ensure continuity for panoramas were previous introduced in StitchDiffusion [3]. Can you please explain whether the methods have significant differences, or they are similar techniques?

Overall, PAR seems to produce high-quality panoramas by adopting masked generation framework with several techniques to ensure panoramic properties. However, there are multiple uncertain points regarding its novelty and evaluation settings, as mentioned above. I believe these critical points should be addressed to clearly understand the advantage of PAR over existing works.

[1] DiffPano: Scalable and Consistent Text to Panorama Generation with Spherical Epipolar-Aware Diffusion, NeurIPS 2024

[2] Cubediff: Repurposing diffusion-based image models for panorama generation, ICLR 2025

[3] Customizing 360-Degree Panoramas through Text-to-Image Diffusion Models, WACV 2024

---

> ### Author Rebuttal · Authors · 2025-07-31
>
> Thank you for recognizing our strengths: 1. Clearly written paper. 2. Reasonable design choice. 3. Strong performance. 4. Thorough ablation studies. We provide more clarifications below.
>
>
> Q1: The difference with StitchDiffusion regarding the proposed dual-space circular padding strategy.
>
> A1: At a high level, both methods aim to mitigate the inconsistency between the left and right ends of the panorama. However, our proposed dual-space circular padding is **fundamentally different** from StitchDiffusion[1].
>
> 1. From methodology, StitchDiffusion focuses on the denoising model, but our method focuses on VAE. StitchDiffusion pads the latent and then denoises twice on the stitch area, which increases the huge computation cost. On the contrary, the extra cost brought by our method is negligible (please refer to our answer for Reviewer Pih7's Q2).
>
>
> 2. From motivation, StitchDiffusion forces the model to fuse seam areas during inference. On the contrary, our pre-padding makes the model learn semantic consistency during the training phase, and our post-padding achieves pixel-level consistency in the inference stage.
>
>
> 3. From the results, we argue that simply focusing on pixel consistency cannot solve the problem of semantic inconsistency, which is demonstrated in Fig.8 of our main manuscript. Moreover, Semantic inconsistency (houses of different heights and colors) is observed in Fig.7c of StitchDiffusion's manuscript.
>
>
>
> Q2: The reason for choosing the outpainting baselines.
>
> A2: We select two **peer-reviewed works with publicly available code**, AOG-Net[2] and 2S-ODIS[3], as the baselines for the Matterport3D dataset. We choose PanoDiff[4] and Diffusion360[5] as OOD baselines because the model weights trained on the SUN360 dataset are provided in the official repo. Following the reviewer's suggestion, we further reproduced PanoDiff and compared it with our 1.4B model on panorama outpainting task. The training dataset is Matterport3D and the text prompt is set to empty. Our model has advantages in terms of FID metric.
>
> |  |PanoDiff | Ours |
> |---|---|---|
> | FID $\downarrow$ | 65.11 | 41.63 |
>
>
>
> Q3: Comparisons with the methods whose codes are unavailable.
>
> A3: We summarize the code and model availability for the mentioned methods in the table below:
>
> | Model | Training Code | Inference Code | Model Weights |
> |---|---|---|---|
> | Diffusion360 | NA | Available | Available |
> | DiffPano | NA | NA | NA |
> | CubeDiff | NA | NA | NA |
>
> As shown, both DiffPano[6] and CubeDiff[7] do not provide publicly available code or pretrained model weights, which makes direct quantitative comparison challenging. Despite this limitation, we have carefully reproduced the DiffPano pipeline according to their paper and compared with our 1.4B model on text-to-panorama task. The training dataset is Matterport3D. The results are as follows, and our model has advantages in terms of FID metric.
>
> | Model | DiffPano | Ours |
> |---|---|---|
> | FID $\downarrow$ | 53.29 | 37.37 |
>
>
>
> Q4: Visualizations of perspective projections.
>
> A4: In Tab.2, our method outperforms baseline methods on the FID-h metric. For FID-h, we horizontally sample 8 perspective images with 90-degree FoV for each panorama and then calculate their FID. As additional visualization results are not permitted in the rebuttal stage, we will add them in the next draft.
>
>
>
> Q5: The overuse of abbreviations seems to harm its readability.
>
> A5: We apologize for the inconvenience. The abbreviations for tasks and metrics are introduced in L78-79 and L207-213, respectively. We will try to make the explanations more explicit in the next draft, including using full names where possible, or providing a brief introduction to the abbreviations in the captions.
>
>
>
> Q6: Visual overview of the proposed framework. How the input and output of the "Transformer f" are structured.
>
> A6: Thanks for your suggestion. Our "Transformer f" follows NOVA[8] and is described in Sec.3.3. Briefly, it takes patchified visual tokens (from the VAE encoder), masking indicators, and text embeddings as inputs. It outputs a conditional signal to drive the subsequent denoising network MLP. We will add a visual overview of the framework in the next draft.
>
>
> [1] Customizing 360-Degree Panoramas through Text-to-Image Diffusion Models, WACV 2024.
>
> [2] Autoregressive Omni-Aware Outpainting for Open-Vocabulary 360-Degree Image Generation, AAAI 2024.
>
> [3] 2S-ODIS: Two-Stage Omni-Directional Image Synthesis by Geometric Distortion Correction, ECCV 2024.
>
> [4] 360-Degree Panorama Generation from Few Unregistered NFoV Images, ACM MM 2023.
>
> [5] Diffusion360: Seamless 360 Degree Panoramic Image Generation based on Diffusion Models, Arxiv 2023.
>
> [6] DiffPano: Scalable and Consistent Text to Panorama Generation with Spherical Epipolar-Aware Diffusion, NeurIPS 2024.
>
> [7] Cubediff: Repurposing diffusion-based image models for panorama generation, ICLR 2025.
>
> [8] Autoregressive Video Generation without Vector Quantization, ICLR 2025.

---

> > ### Author Response · Authors · 2025-08-04
> > **Please let us know whether we address all the issues**
> >
> > Dear reviewer,
> >
> > Thank you for the comments on our paper.
> >
> > We have submitted the response to your comments. Please let us know if you have additional questions so that we can address them during the discussion period. We hope that you can consider raising the score after we address all the issues.
> >
> > Thank you

---

> > ### Comment · Reviewer_1sbD · 2025-08-06
> >
> > I appreciate the authors for providing clarifications on the raised concerns. While my concerns are partially resolved, multiple concerns regarding the method's novelty and the comparison with previous methods yet remain unresolved.
> >
> > **[Q1, A1]** The third point made by the authors is not convincing, since a simple comparison with different settings in different works cannot convey a fair comparison. I believe a systematic evalution or ablation, regarding the existing approaches for panorama padding should have been explored in the original manuscript to support such claims.
> >
> > **[Q2, A2]** The additional experiments provided are not very helpful in understanding the advantage of PAR, because (1) the evaluation only contains PanoDiff while the authors discussed both PanoDiff and Diffusion360, and (2) the evaluation metrics used in the manuscript are missing, and only FID is used.
> >
> > [1] Customizing 360-Degree Panoramas through Text-to-Image Diffusion Models, WACV 2024.

---

> > > ### Author Response · Authors · 2025-08-08
> > > **Further clarification on Q1, Q2**
> > >
> > > Thank you for your detailed and insightful feedback. We appreciate the opportunity to further clarify our contributions and address your concerns.
> > >
> > > ---
> > >
> > > ### 1. **Novelty of Our Approach**
> > >
> > > We would like to emphasize that the **novelty of our work extends beyond the proposed dual-space circular padding strategy**. Our main contributions are as follows:
> > >
> > > - **First application of masked autoregression in panorama generation:**
> > >  This addresses the fundamental conflict between ERP and the i.i.d. assumption in diffusion models. Our approach also enables seamless integration of both text- and image-conditioned generation within a unified architecture.
> > >
> > > - **Translation consistency loss:**
> > >  We introduce a novel loss function to enhance the quality of generated panoramas.
> > >
> > > - **Dual-space padding strategy:**
> > >  Our method improves boundary consistency with negligible computational overhead.
> > >
> > > - **Strong empirical performance and scalability:**
> > >  Our models demonstrate competitive results across various tasks and exhibit promising scalability and generalization.
> > >
> > > ---
> > >
> > > **Due to time and computational resource constraints, we are unable to reproduce additional baselines for whose training code is unavailable during the discussion phase**. For StitchDiffusion and Diffusion360, we conducted zero-shot inference and took great care to ensure fair experimental settings.
> > >
> > > ---
> > >
> > > ### 2. **Comparison with StitchDiffusion**
> > >
> > > **We appreciate the reviewer's recognition of the methodological distinctions we outlined.** We now provide further **quantitative experimental evidence** for the third point of A1.
> > >
> > > We evaluate the discontinuity of generated panoramas for both our method and StitchDiffusion. Specifically, we use our 0.3B model and StitchDiffusion to generate images for 100 textual prompts sampled from a subset of the SUN360 dataset (**excluded from both our and StitchDiffusion's training sets for fairness**), and compute the Discontinuity Score (DS):
> > >
> > > |  | StitchDiffusion | Ours |
> > > |---|---|---|
> > > | DS $\downarrow$ | 1.12 | 0.63 |
> > >
> > > These results indicate that our method achieves significantly better boundary continuity.
> > >
> > > ---
> > >
> > > ### 3. **Comparison with Diffusion360**
> > >
> > > Since the training code for Diffusion360 is unavailable, we performed a zero-shot outpainting comparison on 100 images from the Structured3D dataset (**excluded from both training sets for fairness**). We compare our 1.4B model and Diffusion360. The results are as follows:
> > >
> > > |  | FID $\downarrow$ |  FID-h $\downarrow$ |
> > > |---|---|---|
> > > | Diffusion360 | 140.91| 74.89 |
> > > | Ours | 127.01   | 68.27  |
> > >
> > > Our method consistently outperforms Diffusion360 on both FID and FID-h metrics.
> > >
> > > ---
> > >
> > > ### 4. **Additional Experimental Results**
> > >
> > > To address concerns regarding evaluation metrics and baseline coverage, we further evaluate several baselines using more metrics. **Experimental settings match those in the rebuttal.**
> > >
> > > |  | FAED $\downarrow$ | FID $\downarrow$ | CS $\uparrow$ | DS $\downarrow$ |
> > > |---|---|---|---|---|
> > > | DiffPano | 10.03 | 53.29 | 30.31 | 6.16 |
> > > | Ours | 3.75  | 37.37  | 30.41  | 0.58 |
> > >
> > >
> > > |  | FID $\downarrow$ |  FID-h $\downarrow$ |
> > > |---|---|---|
> > > | PanoDiff | 65.11 | 46.58 |
> > > | Ours | 41.63   | 25.97  |
> > >
> > > We believe these comprehensive results across multiple metrics further demonstrate the strengths of our approach.
> > >
> > > ---
> > >
> > > Thanks again for your thoughtful review. We hope these additional experimental results and clarifications address your concerns. We remain open to further discussion.

---

> > > > ### Comment · Reviewer_1sbD · 2025-08-08
> > > >
> > > > I appreciate the authors for providing clarifications and experiment results regarding the raised concerns. I have carefully read the comments and checked the additional results. The remaining concerns are resolved now, and therefore I would like to raise my initial rating to **borderline accept (4)**.
> > > >
> > > > Please include the above clarifications and detailed comparisons with StitchDiffusion, DiffPano and Diffusion360 in the final version.

---

> > > > > ### Author Response · Authors · 2025-08-08
> > > > >
> > > > > Thanks for your constructive comments and for raising your rating. We will incorporate the clarifications and make the necessary revisions to further improve the quality of our manuscript.

---

### Official Review · Reviewer_Pih7 · 2025-07-01

**Clarity:** 3
**Significance:** 3
**Originality:** 3
**Rating:** 4
**Confidence:** 4

**Summary:**

This paper proposes a panoramic autoregressive model which integrates text and image-conditioned generation within a single architecture. In addition, it proposes specialized designs, including dual-space circular padding and a translation consistency loss.

**Questions:**

The same as weaknesses:
1. add inference time comparison: PAR model vs. other models time comparison

2.circular padding increases the image size and increases the inference cost, can you also show the time comparison in ablation study?

3.Outdoor generalization result is poor, and so is there any plan about improving outdoor generation result.

**Ethical Concerns:**

["NO or VERY MINOR ethics concerns only"]

**Final Justification:**

I have read all the reviews and discussions. The authors have answered all my questions. I will maintain my score (border accept 4).

**Limitations:**

yes.

**Paper Formatting Concerns:**

No.

**Quality:**

3

**Strengths And Weaknesses:**

Strengths:

1.PAR implements unified conditioning   that   separately processes text prompts and image inputs through a singular framework.

2.This paper uses detailed comparative experiments and ablation studies to validate its effectiveness.

Weaknesses:

1.inference time comparison: PAR model vs. other models time comparison

2.circular padding increases the image size and increases the inference cost, can you also show the time comparison in ablation study?

3.Outdoor generalization result is not good without training on outdoor panoramic dataset.

---

> ### Author Rebuttal · Authors · 2025-07-31
>
> Thank you for recognizing our strengths: 1. Unified framework. 2. Detailed experiments and ablation studies. We provide more clarifications below.
>
> Q1: Inference time comparison with other model.
>
> A1: We compare PanFusion[1] with our 0.3B model using one NVIDIA A100 GPU with a batch size of 8. The AR steps are 64, and the denoising steps for MLP are 25.
>
> | Model | PanFusion | Ours |
> |---|---|---|
> | Inference speed (sec/image) $\downarrow$ | 28.91 | 10.03 |
> | FID $\downarrow$ | 45.21 | 41.15 |
>
>
> Q2: Inference time regarding circular padding.
>
> A2: In the inference stage, the circular padding only participates in the VAE decoding phase, which transforms latents into pixels. This means that circular padding does not bring about additional computational cost in the transformer and MLP, which account for the majority of inference time.
>
>
> The table below shows the inference speed for different padding ratios with AR step=16.
>
> | Padding Ratio | 0 (No padding) | 0.0625 | 0.125 | 0.25 | 0.5 |
> |---|---|---|---|---|---|
> | Inference speed (sec/image) $\downarrow$ | 2.99 | 3.01 | 3.02 | 3.04 | 3.05 |
> | Additional costs | 0 | 0.67% | 1.00% | 1.67% | 2.01% |
>
> Please note that:
>
> 1. The proportion of time occupied by circular padding can further decrease as AR steps increase.
>
> 2. From Fig.8 in the main manuscript, the synthesized results with a padding ratio of 0.0625 are close to convergence.
>
>
>
> Q3: Is there any plan to improve the outdoor generation result?
>
> A3: Thanks for your suggestion. Yes, we plan to curate and annotate a large-scale, high-resolution outdoor panoramic dataset with diverse scenes and lighting conditions. This dataset will help reduce the domain gap and provide richer context for model training. Additionally, we are considering incorporating domain-specific data augmentation and leveraging pretraining on related outdoor datasets to enhance generation quality further. We believe these steps will benefit outdoor scene synthesis in future work.
>
>
> [1] Taming Stable Diffusion for Text to 360 Panorama Image Generation, CVPR 2024.

---

> > ### Author Response · Authors · 2025-08-04
> > **Please let us know whether we address all the issues**
> >
> > Dear reviewer,
> >
> > Thank you for the comments on our paper.
> >
> > We have submitted the response to your comments. Please let us know if you have additional questions so that we can address them during the discussion period. We hope that you can consider raising the score after we address all the issues.
> >
> > Thank you

---

> > ### Comment · Reviewer_Pih7 · 2025-08-05
> > **official comment of Pih7**
> >
> > Thank you for your detailed response. You have answered all my questions. However, I still maintain my score.

---

> > > ### Author Response · Authors · 2025-08-08
> > >
> > > Thank you for your feedback. We are glad for any further discussion if needed.

---

### Official Review · Reviewer_YePb · 2025-07-02

**Clarity:** 2
**Significance:** 2
**Originality:** 3
**Rating:** 4
**Confidence:** 5

**Summary:**

This paper introduces a novel method for generating panoramic images from various input modalities, including text and other images. The core contribution lies in adapting masked autoregressive modeling to panoramic image generation and integrating inherent panoramic properties, such as cyclic translation consistency, into the training process. A circular padding approach is also employed to prevent discontinuities. The authors validate their approach through experiments on established datasets like Matterport3D and Sun360.

**Questions:**

- Please address the question regarding the blurriness at the poles mentioned in the weaknesses section.
- Could you elaborate on the advantages and disadvantages of autoregressive modeling compared to multi-view diffusion models for panorama generation?

**Ethical Concerns:**

["NO or VERY MINOR ethics concerns only"]

**Final Justification:**

After reading all discussions and reviews. I increase my initial rating from 3 to 4. The authors provided insights in the blurryness issue at the poles and a comparison to A more recent approach.

**Limitations:**

Yes

**Paper Formatting Concerns:**

No Concerns

**Quality:**

2

**Strengths And Weaknesses:**

Strengths:
- Innovative Approach: The application of masked autoregressive modeling for panorama generation is a compelling idea that effectively addresses key challenges in the field.
- Well-Structured and Comprehensive: The paper is well-written, and the experiments, including the chosen baselines and metrics, are plausible.
- Spatial Consistency: Generated panoramas exhibit strong spatial consistency, outperforming several baselines (as shown in Fig. 3).
Effective Translation Equivariance Evaluation: The evaluation of translation equivariance is thorough.
- Strong Generalization: Impressive results on out-of-distribution (OOD) data highlight the model's excellent generalization capabilities.

Weaknesses:

Image Quality and Photorealism:
- Despite achieving spatial consistency, the panoramic images often lack photorealistic quality. Examples include the washing/drying machines in Fig. 3 and the pool table, which contrasts with recent approaches like CubeDiff (Kalischek et al. ICLR 2025) and Generating 3D Worlds (Schwarz et al. arxiv 2025).
- Significant blurry regions are present at the poles in almost all provided results. The origin of this blurriness needs clarification.

Unclear Methodology:
- Denoising MLP: The proposed small denoising MLP is insufficiently explained and lacks comprehensive ablation studies. Detailed information regarding its architecture, noise schedule, and number of denoising steps is required.
- Training: It is unclear whether the encoder and decoder are trained on ERP (Equirectangular Projection) panoramas.
- Patchification for ERP Images: The exact mechanism of patchification (line 156) for ERP images needs further explanation.

---

> ### Author Rebuttal · Authors · 2025-07-31
>
> Thank you for recognizing our strengths: 1. Innovative approach. 2. The paper is well-written, and the experiments, including the chosen baselines and metrics, are plausible. 3. Strong spatial consistency. 4. Strong Generalization. We provide more clarifications below.
>
> Q1: Lack photorealistic quality compared with CubeDiff and 3D Worlds.
>
> A1: First, as the code and models of CubeDiff[1] and 3D Worlds[2] are unavailable, we carefully check the visualization results in their manuscript and project page. **Incorrect generation results also exist**. Examples include the unrealistic desk (CubeDiff project page, Living Room), twisted table (CubeDiff project page, Alice 2), peach tree on the wall (3D Worlds, Fig.1), and blurred and distorted walls (3D Worlds, Fig.1).
>
> Second, CubeDiff and Generating 3D Worlds focus on the outpainting task, in which the ground truth is projected into the center of the panoramic image, which can easily produce a more realistic feeling to human eyes.
>
> Third, the training data volume. We use the Matterport3D dataset, which contains 9k images for training. On the contrary, CubeDiff collects around 48k images from multiple data sources, and 3D Worlds is trained on DL3DV-10K, which includes 51.2 million frames.
>
>
> Q2: Blurry regions are present at the poles.
>
> A2: Blurry regions at the poles is **an inherent property of the Matterport3D dataset**. In other words, all ground truth images in the Matterport3D dataset have blurriness near the poles. This phenomenon is also visible in the visualization results of PanFusion[3].
>
>
>
> Q3: Details and ablations of the denoising MLP.
>
> A3: We follow NOVA[4] for the construction of the denoising MLP. The MLP has multiple stacked blocks. Within each block, the input is first processed by AdaLNZero to get gating coefficients. It is then transformed using a projection layer consisting of two linear layers with silu activation. The result is then LN-normalized and multiplied by the gating coefficients, and finally combined with a residual connection. We use the Euler solver with 25 denoising steps.
>
> Here is the ablation study for different denoising steps with our 0.3B model. Few steps lead to integral approximation errors, while too many steps lead to cumulative errors.
>
> | Denoising step | 10 | 20 | 30 | 40 |
> |---|---|---|---|---|
> | FID $\downarrow$ | 41.57 | 40.19 | 41.98 | 43.77 |
>
>
>
>
> Q4: Whether the encoder and decoder are trained on ERP panoramas.
>
> A4: We use the default VAE encoder and decoder from NOVA **without any specific fine-tuning**. During the training phase, VAE parameters are frozen, only transformer and MLP parameters are trainable.
>
>
> Q5: The mechanism of patchification in L156.
>
> A5: We employ the patchification operation similar to ViT [5]. Specifically, after obtaining the latent feature maps from the VAE encoder, we apply a 2x2 convolution with a stride of 2. This operation converts spatial feature maps into a sequence of tokens, serving as an input for the subsequent transformer encoder.
>
>
>
> Q6: Discussions on the advantages and disadvantages of autoregressive modeling compared to multi-view diffusion models for panorama generation.
>
> A6: Advantages:
>
> 1. Computational Efficiency:
> Multi-view diffusion models require generating several perspective images (e.g., six views of 512x512 to cover a 512x1024 panorama), resulting in around 9 times computational cost due to self-attention.
>
> 2. Global Consistency:
> Coordinating multiple views demands complex attention mechanisms to ensure coherence, and struggles to maintain a consistent global understanding — sometimes leading to duplicated or inconsistent content.
>
> 3. Fewer Seam Issues:
> Multi-view approaches must handle more seams at the boundaries between views, which increases the difficulty of achieving seamless stitching and can introduce visible artifacts. AR modeling generates the panorama as a whole, inherently avoiding these seam problems.
>
>
> Disadvantages:
>
> Since our model directly generates the entire panoramic image, it needs to use panoramic data to drive the pre-trained model (trained on perspective images) to adapt to the panoramic domain knowledge, which might be less efficient than multi-view modeling because it directly handles perspective images.
>
> Moreover, research in AR-based panorama generation is still limited, and optimal solutions for this task remain to be discovered. We hope our work can inspire further progress in this direction.
>
> We will fruther include these comments in the refined draft.
>
> [1] CubeDiff: Repurposing Diffusion-Based Image Models for Panorama Generation, ICLR 2025.
>
> [2] A Recipe for Generating 3D Worlds From a Single Image, Arxiv 2025.
>
> [3] Taming Stable Diffusion for Text to 360 Panorama Image Generation, CVPR 2024.
>
> [4] Autoregressive Video Generation without Vector Quantization, ICLR 2025.
>
> [5] An Image is Worth 16x16 Words: Transformers for Image Recognition at Scale, ICLR 2021.

---

> > ### Author Response · Authors · 2025-08-04
> > **Please let us know whether we address all the issues**
> >
> > Dear reviewer,
> >
> > Thank you for the comments on our paper.
> >
> > We have submitted the response to your comments. Please let us know if you have additional questions so that we can address them during the discussion period. We hope that you can consider raising the score after we address all the issues.
> >
> > Thank you

---

### Official Review · Reviewer_mEAi · 2025-07-04

**Clarity:** 3
**Significance:** 3
**Originality:** 3
**Rating:** 4
**Confidence:** 4

**Summary:**

This paper proposes Panoramic AutoRegressive (PAR) model which is a unified framework for conditional panoramic image generation, addressing two limitations in existing approaches. The first one is the fundamental incompatibility between diffusion models and equirectangular projection panoramas, which violate the i.i.d. Gaussian noise assumption due to spherical distortions. The second is the unification of text-to-panorama and panorama outpainting tasks within a single autoregressive architecture, eliminating the need for task-specific models or data engineering. PAR leverages masked autoregressive modeling to enable flexible generation orders and integrates two novel components: dual-space circular padding which is applied in latent and pixel spaces to enforce spatial coherence across equirectangular projection boundaries, and a translation consistency loss to provide alignment of the model with panoramic spherical geometry priors. Experiments conducted on Matterport3D demonstrate competitive results (e.g., 37.37 FID for T2P) and strong generalization to out-of-distribution data (SUN360) and zero-shot editing tasks.

**Questions:**

The following actions may improve the paper
1. Add failure cases analysis
2. Add comparison with multitask model Omni^2
3. Conduct computational cost estimation experiments to evaluate training/inference time

**Ethical Concerns:**

["NO or VERY MINOR ethics concerns only"]

**Final Justification:**

I will remain my score. The authors answered the questions and agreed with the comments

**Limitations:**

Yes

**Paper Formatting Concerns:**

No paper formatting concerns detected

**Quality:**

3

**Strengths And Weaknesses:**

Strengths

The paper addresses a fundamental problem in diffusion-based panorama generation masked autoregressive modelling, which provides a theoretically grounded solution. The proposed innovations include dual-space circular padding and translation consistency loss which effectively tackle panoramic-specific challenges like boundary discontinuities and geometric distortions. The proposed PAR method elegantly unifies T2P, PO, and editing tasks within one autoregressive framework, reducing complexity compared to task-specific pipelines (e.g., Omni², which relies on meticulously designed joint training datasets to align heterogeneous tasks, inevitably introducing complexity and domain bias). PAR achieves SOTA FID (37.37) for text-to-panorama generation on Matterport3D dataset, outperforming larger diffusion models (PanFusion: 45.21 FID). It excels in outpaiting task (32.68 FID) and shows compelling qualitative results, avoiding artifacts/repetition in baselines. PAR laos shows strong zero-shot performance on OOD data (SUN360) and editing tasks.

Weaknesses

Text fidelity seems to be a room for performance according to the CLIP scores comparing to PanorLiama. Omni² is only discussed conceptually but not presented in the experimental stage, however when reading I was waiting to see such an experiment. The training/inference costs are not evaluated and presented in the paper, making practical trade-offs unclear. Autoregressive advantave is overemphasized when describing "avoiding i.i.d. assumptions", because it is inherent to the class of autoregressive models.

Clarity. The text is well-structured and mostly clear. The tables and figures are clear and easy to follow.

Significance. The proposed approach addresses critical problems in panorama images generation, which are with practical and useful in VR/AR, robotics and other applications. The code/data release commitment enhances reproducibility.

Originality. The dual-space circular padding and translation consistency loss are novel and well-motivated. Several tasks unification via masked autoregressive modeling is a widely-used technique, but not for this type of data, which can also be stated as an emergency property of the unified architecture.

---

> ### Author Rebuttal · Authors · 2025-07-31
>
> Thank you for recognizing our strengths: 1. The proposed dual-space circular padding and translation consistency loss are novel and well-motivated. 2. The text is well-structured. The tables and figures are clear and easy to follow. 3. Strong performance. We provide more clarifications below.
>
> Q1: Failure cases analysis may improve the paper.
>
> A1: Thanks for your suggestions. Since visualization results cannot be provided in the rebuttal stage, we will add the failure cases analysis in the next draft. According to the visualizations in the main manuscript, the generated images may present unrealistic results in some parts, and there is still a gap compared with real photos, such as some details and textures. This is a common problem of generative models. We believe that scaling data can further improve the generation quality, and we leave it as future work.
>
> Q2: Comparison with Omni2.
>
> A2: As the data, code, and models of Omni2[1] are unavailable, we compare Omni2 with our PAR from two perspectives: dataset construction and modeling. For dataset construction, Omni2 requires complex data engineering, including the collection of over 60k images for various tasks, the application of GroundedSAM and LAMA, with multiple rounds of manual selection for data filtering. On the contrary, we use Matterport3D without specific data preprocessing.
>
> From a modeling perspective, multi-view generation (Omni2 utilizes six perspective views of a cube) presents the following challenges:
>
> 1. Higher computational cost. Using six 512x512 perspective views of a cube instead of a single 512x1024 panorama results in approximately nine times the computational overhead due to the self-attention mechanism.
>
> 2. Poorer global understanding. Due to the existence of multiple views, the model is prone to duplication and requires more complex mechanisms to achieve harmony across the entire scene.
>
> 3. More seam issues. Multi-view approaches must handle more seams at the boundaries between views, which increases the difficulty of achieving seamless stitching and can introduce visible artifacts.
>
>
> Q3: Training and inference time.
>
> A3: Training time: It takes about 18h to train our 0.3B model, and 2 days for the 1.4B model on a single node with 8 NVIDIA A100-80G GPUs.
>
> Inference tine: We benchmark the inference speed for the 0.3B model with different autoregressive steps using one NVIDIA A100 GPU with a batch size of 8. The denoising steps for MLP are 25.
>
> | AR steps | 16 | 32 | 64 |
> |---|---|---|---|
> | Inference speed (sec/image) $\downarrow$ | 3.02 | 5.49 | 10.03 |
>
>
> Q4: CLIP score compared to PanoLlama.
>
> A4: Our model is slightly inferior to PanoLlama in terms of CLIP score, which we argue is due to the gap in panoramic geometry and CLIP's domain knowledge. In Fig.3, the generated results of PanoLlama are more similar to perspective images, while ignoring the panoramic geometry. Since CLIP is pre-trained on massive perspective images, it might have a preference for perspective views.
>
> [1] Omni2: Unifying Omnidirectional Image Generation and Editing in an Omni Model, ACM MM 2025.

---

> > ### Author Response · Authors · 2025-08-04
> > **Please let us know whether we address all the issues**
> >
> > Dear reviewer,
> >
> >
> > Thank you for the comments on our paper.
> >
> >
> > We have submitted the response to your comments. Please let us know if you have additional questions so that we can address them during the discussion period. We hope that you can consider raising the score after we address all the issues.
> >
> >
> > Thank you

---

> > ### Comment · Reviewer_mEAi · 2025-08-05
> >
> > Thanks for the author's reply. I appreciate comparison with Omni2 and detailed answers to my questions. However, I will remain the score

---

> > > ### Author Response · Authors · 2025-08-08
> > >
> > > Thank you for considering our responses. If you have any further questions or concerns, please feel free to let us know.

---

### Note · Authors · 2025-08-13

**Dear Reviewers, ACs, and SACs,**

Thank you for your time and thoughtful feedback on our manuscript. We are grateful for the constructive comments and are pleased that the key concerns have been addressed during the discussion phase.

---

**We appreciate the reviewers for highlighting the strengths of our work:**

- Clear motivation for applying MAR to panorama generation.
- Novel translation consistency loss and the dual-space circular padding strategy.
- Strong empirical performance with promising generalization.
- Well-written manuscript with detailed experiments and ablation studies.

---

**In response to the reviewers' concerns, we provide the following clarifications, which have been acknowledged and will be incorporated into the revised manuscript:**

- **Time analysis:** We now provide detailed analyses of training and inference times. Ablations on the circular padding strategy further demonstrate its efficiency.
- **Comparisons with more baselines:** We include comparisons with recent work (UniPano, ICCV 2025) and several methods with unavailable training code (DiffPano, Diffusion360, StitchDiffusion), demonstrating the advantages of our approach.
- **Ablations on additional datasets:** We have trained our model on the Structured3D dataset for clear pole generation, with advantages over existing AR-based models.
- **Presentation:** We enhance the manuscript with more detailed architecture illustrations and extended discussions on AR-based and multi-view diffusion-based methods.

---

Finally, we confirm that we will release our code and model to facilitate further research and benefit the community. Thank you again for your valuable comments and for helping us improve our work.

**Best regards,**

Authors

---

### Decision · Program_Chairs · 2025-09-17

**Decision:**

Accept (poster)

**Comment:**

This paper adapts masked autoregressive modelling to panoramic images, enabling panorama generation based on different conditions, including in particular text prompts (text-to-panorama) and perspective images (outpainting). Initially the reviews were mixed, criticisms included limited visual quality of the results, including blur near the poles, as well as the absence of certain important baselines. During the discussion the authors made a big effort and were able to address the criticisms, such that the four reviewers converged to a unanimous "borderline accept" rating. The AC agrees that, despite the unresolved limitations, the novelty of adapting masked autoregressive modelling to the task, combined with the breadth of experimental results, warrant acceptance.

[side comment from the AC: it is not good practice to submit a premature paper and only fix the loose ends for the rebuttal. E.g., if you are aware that the poles are blurry because of artefacts in Matterport3D, you should *first* run experiments on another dataset without that problem to verify that it is a dataset issue, *then* submit the paper. Similarly, "their quality is better because they train on more data" is not a very good justification - as long as the additional data is available openly at no cost, it constitutes the current state-of-the-art testbed for research and use should also use it (possibly with an ablation study about dataset size)]